# GANSpace: Discovering Interpretable GAN Controls

**Erik Härkönen**[1,2]        Aaron Hertzmann[2]        Jaakko Lehtinen[1,3]        Sylvain Paris[2]

[1]Aalto University        [2]Adobe Research        [3]NVIDIA

## Abstract

This paper describes a simple technique to analyze Generative Adversarial Networks (GANs) and create interpretable controls for image synthesis, such as change of viewpoint, aging, lighting, and time of day. We identify important latent directions based on Principal Component Analysis (PCA) applied either in latent space or feature space. Then, we show that a large number of interpretable controls can be defined by layer-wise perturbation along the principal directions. Moreover, we show that BigGAN can be controlled with layer-wise inputs in a StyleGAN-like manner. We show results on different GANs trained on various datasets, and demonstrate good qualitative matches to edit directions found through earlier supervised approaches.

## 1 Introduction

Generative Adversarial Networks (GANs) [8], like BigGAN [5] and StyleGAN [10, 11], are powerful image synthesis models that can generate a wide variety of high-quality images, and have already been adopted by digital artists [2]. Unfortunately, such models provide little direct control over image content, other than selecting image classes or adjusting StyleGAN's style vectors. Current attempts to add user control over the output focus on supervised learning of latent directions [9, 7, 25, 19, 16], GAN training with labeled images [12, 22, 21]. However, this requires expensive manual supervision for each new control to be learned. A few methods provide useful control over spatial layout of the generated image [14, 26, 4, 3], provided a user is willing to paint label or edge maps.

This paper shows how to identify new interpretable control directions for existing GANs, without requiring post hoc supervision or expensive optimization: rather than setting out to find a representation for particular concepts ("show me your representation for smile"), our exploratory approach makes it easy to browse through the concepts that the GAN has learned. We build on two main discoveries. First, we show that important directions in GAN latent spaces can be found by applying Principal Component Analysis (PCA) in latent space for StyleGAN, and feature space for BigGAN. Second, we show how BigGAN can be modified to allow StyleGAN-like layer-wise style mixing and control, without retraining. Using these ideas, we show that layer-wise decomposition of PCA edit directions leads to many interpretable controls. Identifying useful control directions then involves an optional one-time user labeling effort.

These mechanisms are algorithmically extremely simple, but lead to surprisingly powerful controls. They allow control over image attributes that vary from straightforward high-level properties such as object pose and shape, to many more-nuanced properties like lighting, facial attributes, and landscape attributes (Figure 1). These directions, moreover, provide understanding about how the GAN operates, by visualizing its "EiGANspace." We show results with BigGAN512-deep and many different StyleGAN and StyleGAN2 models, and demonstrate many novel types of image controls.

One approach is to attempt to train new GANs to be disentangled, e.g., [17]. However, training general models like BigGAN requires enormous computational resources beyond the reach of nearly all potential researchers and users. Hence, we expect that research to interpret and extend the capabilities of existing GANs will become increasingly important.

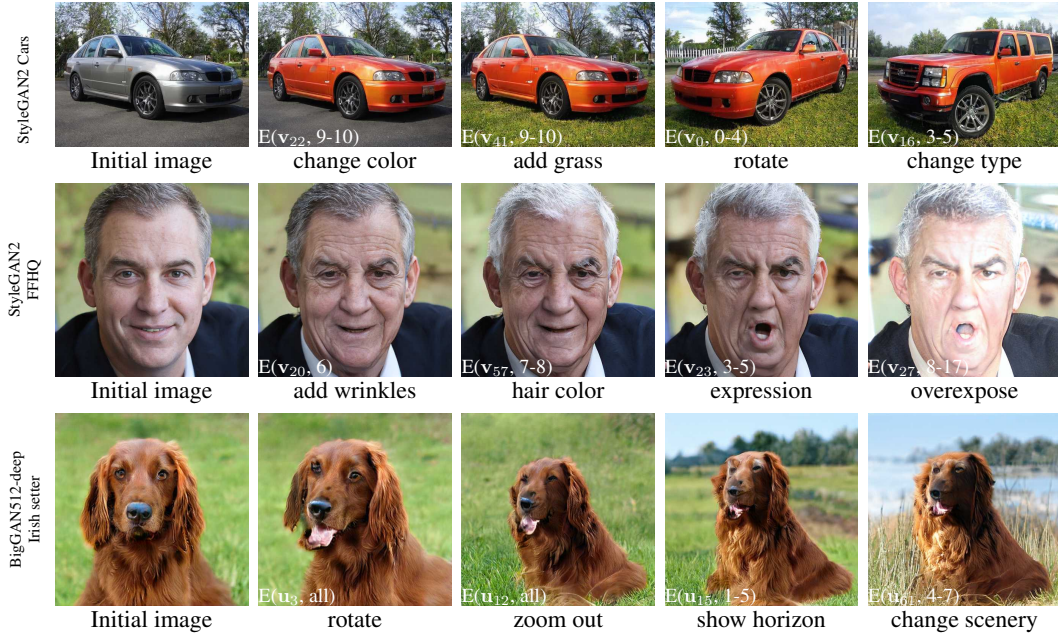

Figure 1: Sequences of image edits performed using control discovered with our method, applied to three different GANs. The white insets specify the edits using notation explained in Section 2.3.

## 2 Discovering GAN Controls

This section describes our new techniques for augmenting existing GANs with new control variables. Our techniques are, algorithmically, very simple. This simplicity is an advantage: for very little effort, these methods enable a range of powerful tools for analysis and control of GANs, that have previously not been demonstrated, or else required expensive supervision. In this paper, we work exclusively with pretrained GANs.

### 2.1 Background

We begin with a brief review of GAN representations [8]. The most basic GAN comprises a probability distribution $p(\mathbf{z})$, from which a latent vector $\mathbf{z}$ is sampled, and a neural network $G(\mathbf{z})$ that produces an output image $I$: $\mathbf{z} \sim p(\mathbf{z})$, $I = G(\mathbf{z})$. The network can be further decomposed into a series of $L$ intermediate layers $G_1...G_L$. The first layer takes the latent vector as input and produces a feature tensor $\mathbf{y}_1 = G_1(\mathbf{z})$ consisting of set of feature maps. The remaining layers each produce features as a function of the previous layer's output: $\mathbf{y}_i = \hat{G}_i(\mathbf{z}) \equiv G_i(\mathbf{y}_{i-1})$. The output of the last layer $I = G_L(\mathbf{y}_{L-1})$ is an RGB image. In the BigGAN model [5], the intermediate layers also take the latent vector as input:

$$\mathbf{y}_i = G_i(\mathbf{y}_{i-1}, \mathbf{z}) \tag{1}$$

which are called Skip-$z$ inputs. BigGAN also uses a class vector as input. In each of our experiments, the class vector is held fixed, so we omit it from this discussion for clarity. In a StyleGAN model [10, 11], the first layer takes a constant input $\mathbf{y}_0$. Instead, the output is controlled by a non-linear function of $\mathbf{z}$ as input to intermediate layers:

$$\mathbf{y}_i = G_i(\mathbf{y}_{i-1}, \mathbf{w}) \qquad \text{with } \mathbf{w} = M(\mathbf{z}) \tag{2}$$

where $M$ is an 8-layer multilayer perceptron. In basic usage, the vectors $\mathbf{w}$ controlling the synthesis at each layer are all equal; the authors demonstrate that allowing each layer to have its own $\mathbf{w}_i$ enables powerful "style mixing," the combination of features of various abstraction levels across generated images.

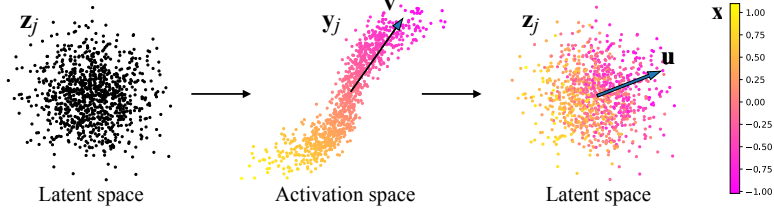

Figure 2: 2D Illustration of identifying a principal activation direction for BigGAN. Random latent vectors $\mathbf{z}_j$ are sampled, and converted to activations $\mathbf{y}_j$. The PCA direction $\mathbf{v}$ is computed from the samples, and PCA coordinates $x_j$ computed, shown here by color-coding. Finally, back in the latent space, the direction $\mathbf{u}$ is computed by regression from $\mathbf{z}_j$ to $x_j$.

## 2.2 Principal Components and Principal Feature Directions

How can we find useful directions in $\mathbf{z}$ space? The isotropic prior distribution $p(\mathbf{z})$ does not indicate which directions are useful. On the other hand, the distribution of outputs in the high-dimensional pixel space is extremely complex, and difficult to reason about. Our main observation is, simply, that the principal components of feature tensors on the early layers of GANs represent important factors of variation. We first describe how the principal components are computed, and then study the properties of the basis they form.

**StyleGAN.** Our procedure is simplest for StyleGAN [10, 11]. Our goal is to identify the principal axes of $p(\mathbf{w})$. To do so, we sample $N$ random vectors $\mathbf{z}_{1:N}$, and compute the corresponding $\mathbf{w}_i = M(\mathbf{z}_i)$ values. We then compute PCA of these $\mathbf{w}_{1:N}$ values. This gives a basis $\mathbf{V}$ for $\mathcal{W}$. Given a new image defined by $\mathbf{w}$, we can edit it by varying PCA coordinates $\mathbf{x}$ before feeding to the synthesis network:

$$\mathbf{w}' = \mathbf{w} + \mathbf{V}\mathbf{x} \tag{3}$$

where each entry $x_k$ of $\mathbf{x}$ is a separate control parameter. The entries $x_k$ are initially zero until modified by a user.

**BigGAN.** For BigGAN [5], the procedure is more complex, because the $\mathbf{z}$ distribution is not learned, and there is no $\mathbf{w}$ latent that parameterizes the output image. We instead perform PCA at an intermediate network layer $i$, and then transfer these directions back to the $\mathbf{z}$ latent space, as follows. We first sample $N$ random latent vectors $\mathbf{z}_{1:N}$; these are processed through the model to produce $N$ feature tensors $\mathbf{y}_{1:N}$ at the $i$th layer, where $\mathbf{y}_j = \hat{G}_i(\mathbf{z}_j)$. We then compute PCA from the $N$ feature tensors, which produces a low-rank basis matrix $\mathbf{V}$, and the data mean $\boldsymbol{\mu}$. The PCA coordinates $\mathbf{x}_j$ of each feature tensor are then computed by projection: $\mathbf{x}_j = \mathbf{V}^T(\mathbf{y}_j - \boldsymbol{\mu})$.

We then transfer this basis to latent space by linear regression, as follows. We start with an individual basis vector $\mathbf{v}_k$ (i.e., a column of $\mathbf{V}$), and the corresponding PCA coordinates $x_{1:N}^k$, where $x_j^k$ is the scalar $k$-th coordinate of $\mathbf{x}_j$. We solve for the corresponding latent basis vector $\mathbf{u}_k$ as:

$$\mathbf{u}_k = \arg\min \sum_j \left\| \mathbf{u}_k x_j^k - \mathbf{z}_j \right\|^2 \tag{4}$$

to identify a latent direction corresponding to this principal component (Figure 2). Equivalently, the whole basis is computed simultaneously with $\mathbf{U} = \arg\min \sum_j \|\mathbf{U}\mathbf{x}_j - \mathbf{z}_j\|^2$, using a standard least-squares solver, without any additional orthogonality constraints. Each column of $\mathbf{U}$ then aligns to the variation along the corresponding column of $\mathbf{V}$. We call the columns $\mathbf{u}_k$ *principal directions*. We use a new set of $N$ random latent vectors for the regression. Editing images proceeds similarly to the StyleGAN case, with the $x_k$ coordinates specifying offsets along the columns $\mathbf{u}_k$ of the principal direction matrix: $\mathbf{z}' = \mathbf{z} + \mathbf{U}\mathbf{x}$.

We compute PCA at the first linear layer of BigGAN512-deep, which is the first layer with a non-isotropic distribution. We found that this gave more useful controls than later layers. Likewise, for StyleGAN, we found that PCA in $\mathcal{W}$ gave better results than applying PCA on feature tensors and then transferring to latent space $\mathbf{w}$.

Examples of the first few principal components are shown in Figure 3(top) for StyleGAN2 trained on FFHQ; see also the beginning of the accompanying video. While they capture important concepts,

some of them entangle several separate concepts. Similar visualizations are shown for other models (in Section 1 of the Supplemental Material, abbreviated SM §1 later).

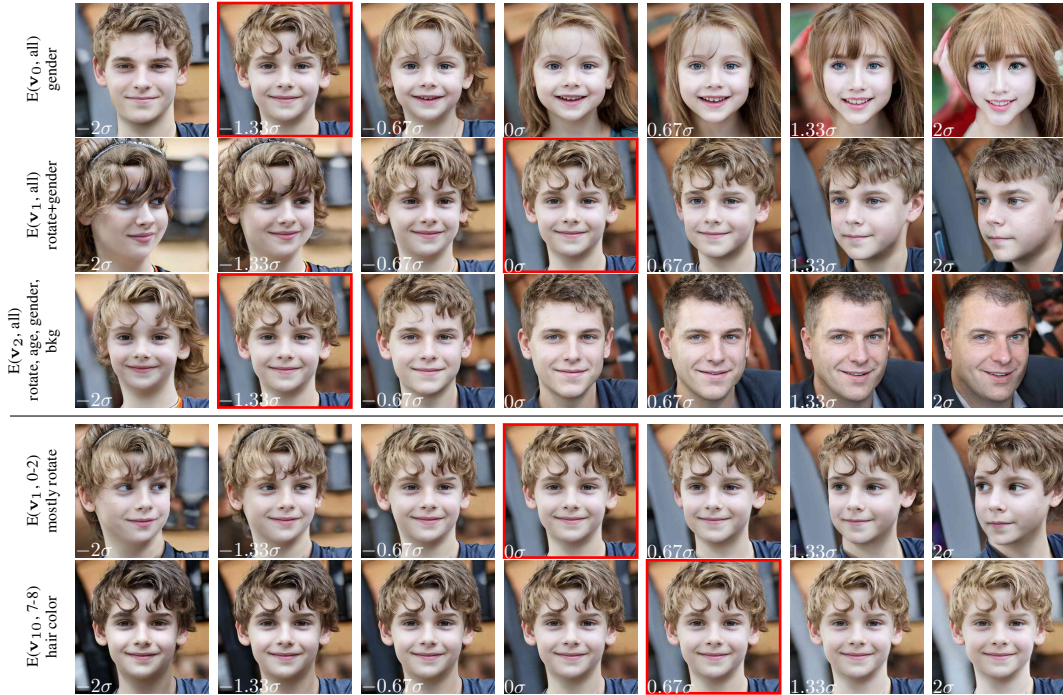

Figure 3: *Rows 1-3* illustrate the three largest principal components in the intermediate $\mathcal{W}$ latent space of StyleGAN2. They span the major variations expected of portrait photographs—such as gender and head rotation—with a few effects typically entangled together. The red square corresponds to location of the original image on each principal axis. *Rows 4-5* demonstrate the effect of constraining the variation to a subset of the layers. For example, restricting the 2nd component to only layers 0-2, denoted $E(\mathbf{v}_1, 0\text{-}2)$, leaves a relatively pure head rotation that changes gender expression and identity less (compare to 2nd row). Similarly, selective application of the principal components allows control of features such as hair color, aspects of hairstyle, and lighting. See SM §1 for a larger sampling.

## 2.3 Layer-wise Edits

Given the directions found with PCA, we now show that these can be decomposed into interpretable edits by applying them only to certain layers.

**StyleGAN.** StyleGAN provides layerwise control via the $\mathbf{w}_i$ intermediate latent vectors. Given an image with latent vector $\mathbf{w}$, layerwise edits entail modifying only the $\mathbf{w}$ inputs to a range of layers, leaving the other layers' inputs unchanged. We use notation $E(\mathbf{v}_i, \text{j-k})$ to denote edit directions; for example, $E(\mathbf{v}_1, 0\text{-}3)$ means moving along component $\mathbf{v}_1$ at the first four layers only. $E(\mathbf{v}_2, \text{all})$ means moving along component $\mathbf{v}_2$ globally: in the latent space and to all layer inputs. Edits in the $\mathcal{Z}$ latent space are denoted $E(\mathbf{u}_i, \text{j-k})$.

This is illustrated in the last rows of Figure 3. For example, component $\mathbf{v}_1$, which controls head rotation and gender in an entangled manner, controls a purer rotation when only applied to the first three layers in $E(\mathbf{v}_1, 0\text{-}2)$; similarly, the age and hairstyle changes associated with component $\mathbf{v}_4$ can be removed to yield a cleaner change of lighting by restricting the effect to later layers in $E(\mathbf{v}_4, 5\text{-}17)$. It is generally easy to discover surprisingly targeted changes from the later principal components. Examples include $E(\mathbf{v}_{10}, 7\text{-}8)$ that controls hair color, as well as $E(\mathbf{v}_{11}, 0\text{-}4)$ that controls the height of the hair above the forehead. More examples across several models are shown in Figure 7; see also the accompanying video. As shown in Figure 1, multiple edits applied simultaneously across multiple principal directions and internal layer ranges compose well.

**BigGAN.** BigGAN does not have a built-in layerwise control mechanism. However, we find that **BigGAN can be modified to produce behavior similar to StyleGAN**, by varying the intermediate Skip-$z$ inputs $\mathbf{z}_i$ separately from the latent $\mathbf{z}$: $\mathbf{y}_i = G(\mathbf{y}_{i-1}, \mathbf{z}_i)$. Here the latent inputs $\mathbf{z}_i$ are allowed to vary individually between layers in a direct analogy to the style mixing of StyleGAN. By default, all inputs are determined by an initial sampled or estimated $\mathbf{z}$, but then edits may be performed to the inputs to different layers independently. Despite the fact that BigGAN is trained without style mixing regularization, we find that it still models images in a form of style/content hierarchy. Figure 6 shows the effect of transferring intermediate latent vectors from one image to another. Like StyleGAN, transferring at lower layers (closer to the output) yields lower-level style edits. See SM §2 for more examples of BigGAN style mixing. Since the Skip-$z$ connections were not trained for style resampling, we find them to be subjectively "more entangled" than the StyleGAN style vectors. However, they are still useful for layerwise editing, as shown in Figures 7 and SM §1: we discover components that control, for instance, lushness of foliage, illumination and time of day, and cloudiness, when applied to a select range of layers.

**Interface.** We have created a simple user interface that enables interactive exploration of the principal directions via simple slider controls. Layer-wise application is enabled by specifying a start and end layer for which the edits are to be applied. The GUI also enables the user to name the discovered directions, as well as load and save sets of directions. The exploration process is demonstrated in the video, and the runnable Python code is attached as supplemental material.

## 3 Findings and Results

We describe a number of discoveries from our PCA analysis, some of which we believe are rather surprising. We also show baseline comparisons. We show edits discovered on state-of-the-art pretrained GANs, including BigGAN512-deep, StyleGAN (Bedrooms, Landscapes, WikiArt training sets), and StyleGAN2 (FFHQ, Cars, Cats, Church, Horse training sets). Details of the computation and the pretrained model sources are found in SM §3. This analysis reveals properties underlying the StyleGAN and BigGAN models.

### 3.1 GAN and PCA Properties

Across all trained models we have explored, **large-scale changes to geometric configuration and viewpoint are limited to the first 20 principal components ($\mathbf{v}_0$-$\mathbf{v}_{20}$); successive components leave layout unchanged, and instead control object appearance/background and details**. As an example, Figure 3 shows edit directions for the top 3 PCA components in a StyleGAN2 model trained on the FFHQ face dataset [10]. We observe that the first few components control large-scale variations, including apparent gender expression and head rotation. For example, component $\mathbf{v}_0$ is a relatively disentangled gender control; component $\mathbf{v}_1$ mixes head rotation and gender, and so on. See SM §1 for a visualization of the first 20 principal components.

PCA also reveals that **StyleGANv2's latent distribution $p(\mathbf{w})$ has a relatively simple structure:** the principal coordinates are nearly-independent variables with non-Gaussian unimodal distributions. We also find that **the first 100 principal components are sufficient to describe overall image appearance**; the remaining 412 dimensions control subtle though perceptible changes in appearance; see SM §4 and §5 for details and examples.

We find that **BigGAN components appear to be class-independent**, e.g., PCA components for one class were identical to PCA components for another class in the cases we tested. SM §6 shows examples of PCA components computed at the first linear layer of BigGAN512-deep for the husky class. We find that the global motion components have the same effect in different classes (e.g., component 6 is zoom for all classes tested), but later components may have differing interpretations across classes. For instance, a direction that makes the image more blue might mean winter for some classes, but just nighttime for others.

### 3.2 Model entanglements and disallowed combinations

We observe a number of properties of GAN principal components that seem to be inherited from GANs' training sets. In some cases, these properties may be desirable, and some may be limitations

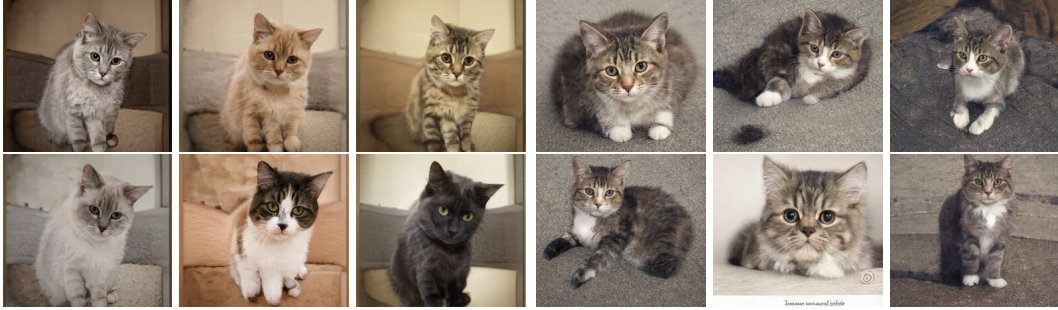

(a) Fix first 8 PCA coord., randomize remaining 504 (Pose and camera fixed, appearance changes)

(b) Randomize first 8 PCA coord., fix remaining 504 (Appearance fixed, pose changes)

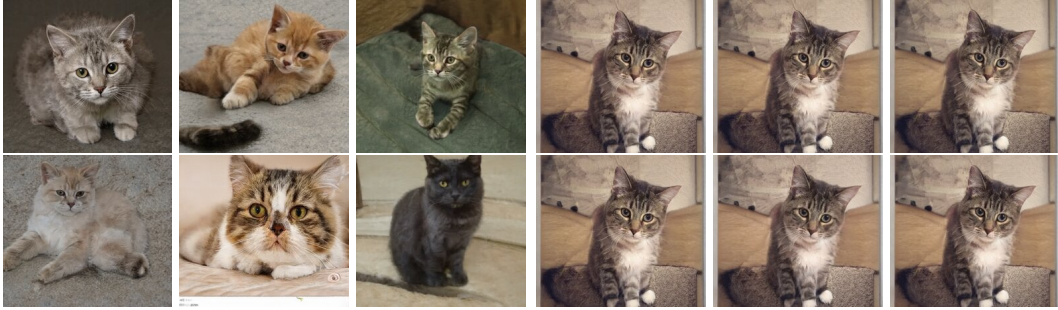

(c) Fix 8 random basis coord., randomize the others (Almost everything changes)

(d) Randomize 8 random basis coord., fix remaining 504 (Almost nothing changes)

Figure 4: Illustration of the significance of the principal components as compared to random directions in the intermediate latent space $\mathcal{W}$ of StyleGAN2. Fixing and randomizing the early principal components shows a separation between pose and style (a, b). In contrast, fixing and randomizing randomly-chosen directions does not yield a similar meaningful decomposition (c, d).

of our approach. Some of these may also be seen as undesirable biases of the trained GAN. Our analysis provides one way to identify these properties and biases that would otherwise be hard to find.

For StyleGAN2 trained on the FFHQ face dataset, geometric changes are limited to rotations in the first 3 components. No translations are discovered, due to the carefully aligned training set.

Even with our layer-wise edits, we observe some entanglements between distinct concepts. For example, adjusting a car to be more "sporty" causes a more "open road" background, whereas a more "family" car appears in woods or city streets. This plausibly reflects typical backgrounds in marketing photographs of cars. Rotating a dog often causes its mouth to open, perhaps a product of correlations in portraits of dogs. For the "gender" edit, one extreme "male" side seems to place the subject in front of a microphone; whereas the "female" side is a more frontal portrait. See SM §7 for examples.

We also observe "disallowed combinations," attributes that the model will not apply to certain faces. The "Wrinkles" edit will age and add wrinkles to adult faces, but has no significant effect on a child's face. Makeup and Lipstick edits add/remove makeup to female-presenting faces, but have little or no effect on male faces. When combining the two edits for "masculine" and "adult," all combinations work, except for when trying to make a "masculine child." See SM §7 for Figures.

## 3.3 Comparisons

No previously published work addresses the problem we consider, namely, unsupervised identification of interpretable directions in an existing GAN. In order to demonstrate the benefits of our approach, we show qualitative comparisons to random directions and supervised methods.

**Random directions.** We first compare the PCA directions to randomly-selected directions in $\mathcal{W}$. Note that there are no intrinsically-preferred directions in this space, i.e., since $\mathbf{z}$ is isotropic, the canonical directions in $\mathbf{z}$ are equivalent to random directions. As discussed in the previous section,

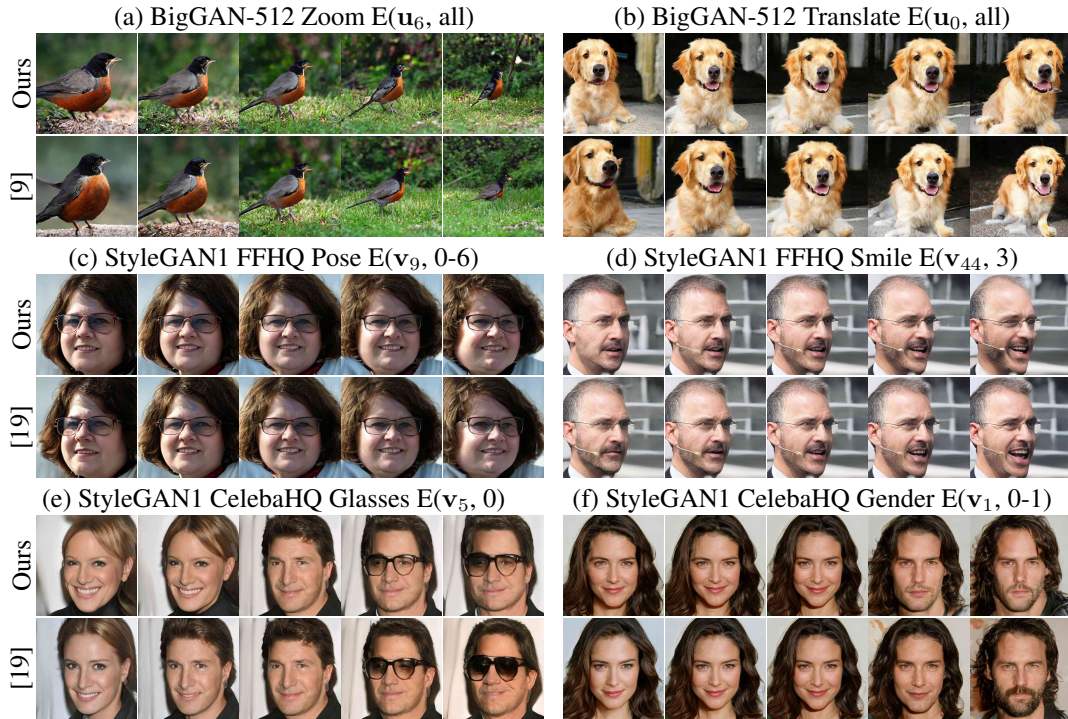

Figure 5: Comparison of edit directions found through PCA to those found in previous work using supervised methods [9, 19]. Some are visually very close (a, c). Others achieve a variant of the same effect (d, e, f), sometimes with more entanglement (d), and sometimes less (b). In some cases, both produce highly entangled effects (a, f). We also observe a few cases where strong effects introduce inconsistencies (e) in our outputs. Still, the results are remarkably close given that our approach does not specify target transformations or use supervised learning. The corresponding edits were found manually using our interactive exploration software.

PCA provides a useful ordering of directions, separating the pose and the most significant appearance into the first components. As illustrated in SM §8, each random direction includes some mixture of pose and appearance, with no separation among them.

We further illustrate this point by randomizing different subsets of principal coordinates versus random coordinates. Figure 4 contains four quadrants, each of which shows random perturbations about a latent vector that is shared for the entire figure. In Figure 4a, the first eight principal coordinates $x_{0...7}$ are fixed and the remaining 504 coordinates $x_{8...512}$ are randomized. This yields images where the cat pose and camera angle are held roughly constant, but the appearance of the cat and the background vary. Conversely, fixing the last 504 coordinates and randomizing the first eight (Figure 4b) yields images where the color and appearance are held roughly constant, but the camera and orientation vary. The bottom row shows the results of the same process applied to random directions; illustrating that any given 8 directions have no distinctive effect on the output. SM §8 contains more examples.

**Supervised methods**    Previous methods for finding interpretable directions in GAN latent spaces require outside supervision, such as labeled training images or pretrained classifiers, whereas our approach aims to automatically identify variations intrinsic to the model without supervision.

In Figure 5, we compare some of our BigGAN zoom and translation edits to comparable edits found by supervised methods [9], and our StyleGAN face attribute edits to a supervised method [19]. In our results, we observe a tendency for slightly more entanglement (for example, loss of microphone and hair in Figure 5d); moreover, variations of similar effects can often be obtained using multiple components. More examples from different latent vectors are shown in SM §8. However, we emphasize that (a) our method obtained these results without any supervision, and (b) we have been able to identify many edits that have not previously been demonstrated; supervising each of

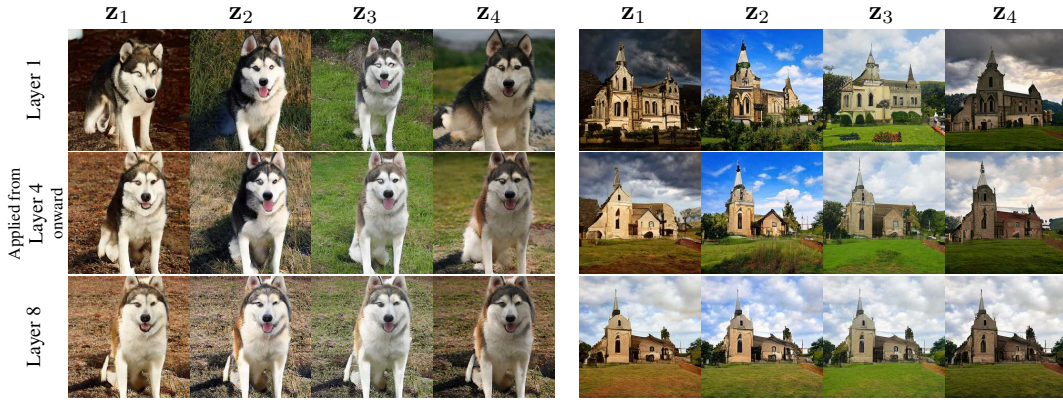

Figure 6: Style variation in BigGAN. Changing the latent vector in BigGAN in the middle of the network alters the style of the generated image. The images on the top row are generated from a base latent (not shown) by substituting $z_1 \ldots z_4$ in its place from layer 1 onwards (resp. from layer 4 and 8 onwards on the following rows). Early changes affect the entire image, while later changes produce more local and subtle variations. Notably, comparing the dog and church images on the last row reveals the latents have class-agnostic effects on color and texture.

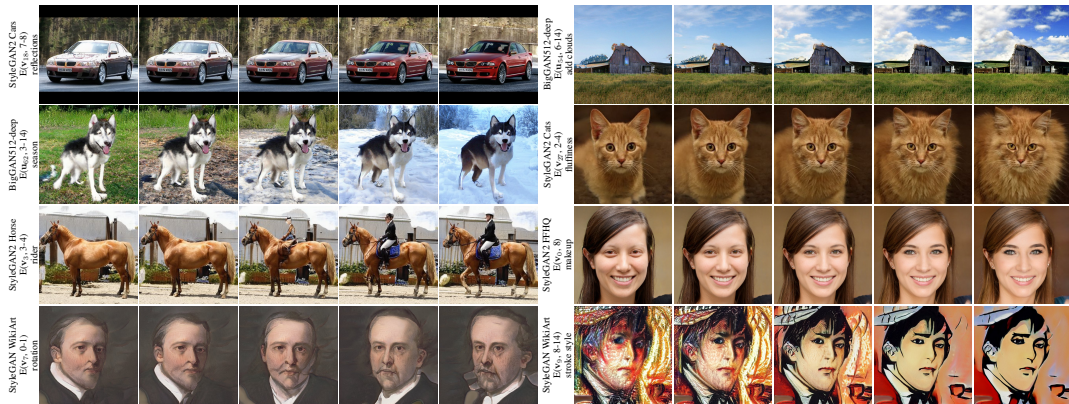

Figure 7: A selection of interpretable edits discovered by selective application of latent edits across the layers of several pretrained GAN models. The reader is encouraged to zoom in on an electronic device. A larger selection is available in SM §1.

these would be very costly, and, moreover, it would be hard to know in advance which edits are even possible with these GANs.

## 4 Discussion

This paper demonstrates simple but powerful ways to create images with existing GANs. Rather than training a new model for each task, we take existing general-purpose image representations and discover techniques for controlling them. This work suggests considerable future opportunity to analyze these image representations and discover richer control techniques in these spaces, for example, using other unsupervised methods besides PCA. Our early experiments with performing PCA on other arrangements of the feature maps were promising. A number of our observations suggest improvements to GAN architectures and training, perhaps similar to [6]. It would be interesting to compare PCA directions to those learned by concurrent work in disentanglement, e.g., [13]. Our approach also suggests ideas for supervised training of edits, such as using our representation to narrow the search space. Several methods developed concurrently to our own explore similar or related ideas [23, 15, 24, 20, 1], and comparing or combining approaches may prove useful as well.

## Broader Impact

As our method is an image synthesis tool, it shares with other image synthesis tools the same potential benefits (e.g., [2]) and dangers that have been discussed extensively elsewhere, e.g., see [18] for one such discussion.

Our method does not perform any training on images; it takes an existing GAN as input. As discussed in Section 3.2, our method inherits the biases of the input GAN, e.g., limited ability to place makeup on male-presenting faces. Conversely, this method provides a tool for discovering biases that would otherwise be hard to identify.

## Acknowledgments and Disclosure of Funding

We thank Miika Aittala for insightful discussions and Tuomas Kynkäänniemi for help in preparing the comparison to Jahanian et al. [9]. Thanks to Joel Simon for providing the Artbreeder Landscape model. This work was created using computational resources provided by the Aalto Science-IT project.

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
