[Supplementary Material]

# Supplementary Material:
# GANSpace: Discovering Interpretable GAN Controls

**Erik Härkönen**[1,2]     Aaron Hertzmann[2]     Jaakko Lehtinen[1,3]     Sylvain Paris[2]

[1]Aalto University     [2]Adobe Research     [3]NVIDIA

## 1   Examples of Principal Components and Layerwise Edits

Figure 1 shows an assortment of interpretable edits discovered with our method for many different models.

We visualize the first 20 Principal Components for several models:StyleGAN2 FFHQ (Figure 2a), StyleGAN2 Cars (Figure 4a), StyleGAN2 Cats (Figure 3a), and BigGAN512-deep Husky (Figure 5a). The images are centered at the mean of each component, which causes slight differences within the center columns.

## 2   BigGAN style mixing

Figure 6 shows a more detailed example of mixing style and content at different layers in BigGAN [1].

## 3   Model and Computation Details

We use incremental PCA [5] for efficient computation, and use $N = 10^6$ samples. On a relatively high-end desktop PC, computation takes around 1.5 hours on BigGAN512-deep and 2 minutes on StyleGAN and StyleGAN2.

Our StyleGAN model weights were obtained from `https://github.com/justinpinkney/awesome-pretrained-stylegan`, except for Landscapes, which was provided by `artbreeder.com`. Our StyleGAN2 models were those provided by the authors online [4].

The sliders in our GUI operate in units of standard deviations, and we find that later components work for wider ranges of values than earlier ones. The first ten or so principal components, such as head rotation (E($\mathbf{v}_1$, 0-2)) and lightness/background (E($\mathbf{v}_8$, 5)), operate well in the range $[-2...2]$, beyond which the image becomes unrealistic. In contrast, face roundness (E($\mathbf{v}_{37}$, 0-4)) can work well in the range $[-20...20]$, when using $0.7$ as the truncation parameter.

For truncation, we use interpolation to the mean as in StyleGAN [3]. The variation in slider ranges described above suggests that truncation by restricting $\mathbf{w}$ to lie within 2 standard deviations of the mean would be a very conservative limitation on the expressivity of the interface, since it can produce interesting images outside this range.

A video showcasing our exploration UI is available at `https://youtu.be/jdTICDa_eAI`. The code of our method is hosted online: `https://github.com/harskish/ganspace`.

## 4   How many components are needed?

We first investigate how many dimensions of the latent space are important to image synthesis. Figure 7 shows the variance captured in each dimension of the PCA for the FFHQ model. The first 100 dimensions capture 85% of the varaince; the first 200 dimensions capture 92.5%, and the first 400 dimensions capture 98.5%.

What does this correspond to visually? Figure 8 shows images randomly sampled, and then projected to a reduced set of PCA components. That is, we sample $\mathbf{w} \sim p(\mathbf{w})$, and then replace it with $\mathbf{w} \leftarrow \mathbf{V}_K \mathbf{V}_K^T (\mathbf{w} - \mu) + \mu$, where $\mathbf{V}_K$ are the columns for the first $K$ principal components. Observe that nearly all overall face details are captured by the first 100 components; the remaining 412 components make small adjustments to shape and tone.

## 5 What is $p(\mathbf{w})$?

Inspecting the marginal distributions of the principal coordinates gives insight as to the shape of $p(\mathbf{w})$, the distribution over latents . In principle, the learned distribution could have any shape, within the range of what can be parameterized by an 8-layer fully-connected network $M(\mathbf{z})$. For example, it could be highly multimodal, with different modes for different clusters of training image. One could imagine, for example, different clusters for discrete properties like eyeglasses/no-eyeglasses, or the non-uniform distribution of other attributes in the training data.

In fact, we find that this is not the case: for all of the StyleGANv2 models, PCA analysis reveals that $p(\mathbf{w})$ has a rather simple form. Through this analysis, we can describe the shape of $p(\mathbf{w})$ very thoroughly. The conclusions we describe here could be used in the future to reduce the dimensionality of StyleGAN models, either during training or as a post-process.

**Sampling.** To perform this analysis, we sample $N = 10^6$ new samples $\mathbf{w}_i \sim p(\mathbf{w})$, and then project them with our estimated PCA basis:

$$\mathbf{x}_i = \mathbf{V}^T (\mathbf{w}_i - \mu) \tag{1}$$

where $\mathbf{V}$ is a full-rank PCA matrix $(512 \times 512)$ for our StyleGAN2 models. We then analyze the empirical distribution of these $\mathbf{x}$ samples.

The experiments described here are for the FFHQ face model, but we have observed similar phenomena for other models.

**Independence.** PCA projection decorrelates variables, but does not guarantee independence; it may not even be possible to obtain linear independent components for the distribution.

Let $x^i$ and $x^j$ be two entries of the $\mathbf{x}$ variable. We can assess their independence by computing Mutual Information (MI) between these variables. We compute MI numerically, using a $1000 \times 1000$-bin histogram of the joint distribution $p(x^{(j)}, x^{(k)})$. The MI of this distribution is denoted $I_{jk}$. Note that the MI of a variable with itself is equal to the entropy of that variable $H_j = I_{jj}$, and both quantities are measured in bits. We find that the entropies lie in the range $H_j \in [6.9, 8.7]$ bits. In contrast, the MIs lie in the range $I_{jk} \in [0, 0.3]$ bits.

This indicates that, empirically, the principal components are very nearly independent, and we can understand the distribution by studying the individual components separately.

**Individual distributions.** What do the individual distributions $p(x^j)$ look like? Figure 9 shows example histograms of these variables. As visible in the plots, the histograms are remarkably unimodal, without heavy tails. Visually they all appear Gaussian, though plotting them in the log domain reveals some asymmetries.

**Complete distribution.** This analysis suggests that the sampler for $\mathbf{w}$ could be replaced with the following model:

$$x^j \sim p(x^j) \tag{2}$$
$$\mathbf{y} = \mathbf{V}\mathbf{x} + \mu \tag{3}$$

where the one-dimensional distributions $p(x^j)$ are in some suitable form to capture the unimodal distributions described above. This is a multivariate distribution slightly distorted from a Gaussian.

This representation would have substantially fewer parameters than the $M(\mathbf{z})$ representation in StyleGAN.

## 6 BigGAN Principal Directions are Class-agnostic

Figure 10 shows examples of transferring edits between BigGAN classes, illustrating our observation that PCA components seem to be the same across different BigGAN classes.

## 7 Entanglements and Disallowed Combinations

Most of our edits work across different starting images in a predictable way. For example, the head rotation edit accurately rotates any head in our tests. However, as discussed in Section 3.2 of the paper, some edits show behavior that may reveal built-in priors or biases learned by the GAN. These are illustrated in Figures 11 ("baldness"), 12 ("makeup"), 13 ("white hair"), and 14 ("wrinkles"): in each case, different results occur when the same edit is applied to difference starting images. Figure 15 shows an example of combining edits, where one combination is not allowed by the model.

## 8 Comparisons

Figures 16, 17, and 18 show comparisons of edits discovered with our method to those discovered by the supervised methods [6] and [2].

Sets of 20 normally distributed random directions $\{\hat{\mathbf{r}}_0 \ldots \hat{\mathbf{r}}_{19}\}$ in $\mathcal{Z}$ are shown for StyleGAN2 FFHQ (Figure 2a), StyleGAN2 Cars (Figure 4b), StyleGAN2 Cats (Figure 3b), and BigGAN512-deep Husky (Figure 5b). The directions are scaled in order to make the effects more visible.

Figures 19, 20, 21, and 22 visualize the significance of the PCA basis as compared to a random basis in latent space.

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

Figure 1: A selection of interpretable edits discovered by selective application of latent edits across the layers of several pretrained GAN models. The reader is encouraged to zoom in on an electronic device.

(a) Principal components $\mathbf{v_0} - \mathbf{v_{19}}$, $\pm 2\sigma$     (b) Normally distributed directions in $\mathcal{Z}$, $\pm 10\hat{r}_i$

Figure 2: A visualization of the first 20 principal components of StyleGAN2 FFHQ (a), and of 20 isotropic Gaussian directions in $\mathcal{Z}$ (b). The random directions are scaled to emphasize their effect.

(a) Principal components $\mathbf{v_0} - \mathbf{v_{19}}$, $\pm 2\sigma$     (b) Normally distributed directions in $\mathcal{Z}$, $\pm 10\hat{r}_i$

Figure 3: A visualization of the first 20 principal components of StyleGAN2 Cats (a), and of 20 isotropic Gaussian directions in $\mathcal{Z}$ (b). The random directions are scaled to emphasize their effect.

(a) Principal components $\mathbf{v_0} - \mathbf{v_{19}}$, $\pm 2\sigma$     (b) Normally distributed directions in $\mathcal{Z}$, $\pm 10\hat{r}_i$

Figure 4: A visualization of the first 20 principal components of StyleGAN2 Cars (a), and of 20 isotropic Gaussian directions in $\mathcal{Z}$ (b). The random directions are scaled to emphasize their effect.

(a) Principal components $\mathbf{u_0} - \mathbf{u_{19}}$, $\pm 2\sigma$   (b) Normally distributed directions in $\mathcal{Z}$, $\pm 6\hat{r}_i$

Figure 5: A visualization of the first 20 principal components of BigGAN512-deep husky (a), and of 20 isotropic Gaussian directions in $\mathcal{Z}$ (b). The random directions are scaled to emphasize their effect.

Figure 6: Even though not explicitly trained to do so, BigGAN displays similar style-mixing characteristics to StyleGAN. Here, the latent vector of the content image is swapped for that of the style image starting at different layers.

Figure 7: Variance of the principal components for StyleGANv2 FFHQ.

Figure 8: Randomly sampled images, projected onto reduced numbers of PCA dimensions: 0, 1, 5, 10, 20, 100, 512 (full dimensional).

Figure 9: Top: Marginal distributions for $x^{(0)}$, $x^{(18)}$, $x^{(20)}$, $x^{(50)}$. Bottom: log domain for these distributions

Figure 10: The latent space directions we discover often generalize between BigGAN classes. *Left three columns:* Component 0 corresponds to translation and component 6 to zoom. The edit is applied globally to all layers. *Right three columns:* Some later components, when applied to a subset of the layers, control specific textural aspects such as clouds or nighttime illumination of a central object. The components shown where all computed from the husky class.

Figure 11: An example of edit direction dependence on input face: StyleGAN FFHQ direction that we labeled as "baldness" ($E(\mathbf{v}_{21}, 2\text{-}4)$).

Figure 12: An example of edit direction dependence on input face: StyleGAN FFHQ direction that we labeled as "makeup" ($E(\mathbf{v}_0, 8)$).

Figure 13: An example of edit direction dependence on input face: StyleGAN FFHQ direction that we labeled as "white hair" ($E(\mathbf{v}_{57}, 7\text{-}9)$).

Figure 14: An example of edit direction dependence on input face: StyleGAN FFHQ direction that we labeled as "wrinkles" ($E(\mathbf{v}_{20}, 6)$).

Figure 15: Combining edits. Starting with the center image, the horizontal axis corresponds to adding or removing elements of $x_0$, in the range $\Delta x_0 \in [-3, 3]$. The horizontal axis is adding/removing elements of $x_{18}$. Note that the horizontal axis roughly corresponds to "masculinity" and the vertical to "age." The components operate independently, except that the model does not produce a "masculine little boy" in the upper-left.

(a) Zoom E($\mathbf{u}_6$, all)

(b) Translate E($\mathbf{u}_0$, all)

Ours

[2]

(a) FFHQ Blueness E($\mathbf{u}_2$, 17)

(b) FFHQ Greenness E($\mathbf{u}_1$, 17)

Ours

[2]

(a) Rotate E($\mathbf{u}_0$, 0)

(b) ShiftY E($\mathbf{u}_7$, 1)

Ours

[2]

Ours

[2]

Figure 16: Comparisons against [2] for BigGAN512-deep, StyleGAN FFHQ, and StyleGAN Cars.

(c) FFHQ Pose E($\mathbf{v}_9$, 0-6)  (d) FFHQ Gender E($\mathbf{v}_0$, 2-5)

Ours

[6]

Ours

[6]

(e) FFHQ Smile E($\mathbf{v}_{44}$, 3)  (f) FFHQ Glasses E($\mathbf{v}_{12}$, 0-1)

Ours

[6]

Ours

[6]

Figure 17: Edits found with our method compared to those found by [6] for the StyleGAN FFHQ model.

(g) CelebaHQ Pose $E(\mathbf{v}_7, 0\text{-}6)$

(h) CelebaHQ Gender $E(\mathbf{v}_1, 0\text{-}1)$

Ours

[6]

Ours

[6]

(i) CelebaHQ Smile $E(\mathbf{v}_{14}, 3)$

(j) CelebaHQ Glasses $E(\mathbf{v}_5, 0)$

Ours

[6]

Ours

[6]

Figure 18: Edits found with our method compared to those found by [6] for the StyleGAN CelebaHQ model

StyleGAN2 car

(a) Fix first 5 PCA coord., randomize rest

(b) Randomize first 5 PCA coord., fix rest

(c) Fix 5 random basis coord., randomize rest

(d) Randomize 5 random basis coord., fix rest

Figure 19: The PCA basis displays a content-style separation not present in random bases.

BigGAN256-deep duck

(a) Fix first 10 PCA coord., randomize rest

(b) Randomize first 10 PCA coord., fix rest

(c) Fix 10 random basis coord., randomize rest

(d) Randomize 10 random basis coord., fix rest

Figure 20: The PCA basis displays a content-style separation not present in random bases.

StyleGAN bedrooms

(a) Fix first 10 PCA coord., randomize rest

(b) Randomize first 10 PCA coord., fix rest

(c) Fix 10 random basis coord., randomize rest

(d) Randomize 10 random basis coord., fix rest

Figure 21: The PCA basis displays a content-style separation not present in random bases.

StyleGAN ffhq

(a) Fix first 10 PCA coord., randomize rest

(b) Randomize first 10 PCA coord., fix rest

(c) Fix 10 random basis coord., randomize rest

(d) Randomize 10 random basis coord., fix rest

Figure 22: The first few principal components often encode style changes in addition to geometry in spatially aligned datasets, as seen in the change of identity in the top right quadrant.