[Reviews · NeurIPS 2020]

Review 1

Summary and Contributions: The paper argues that PCA directions in the space of StyleGAN/BigGAN activations often correspond to human-interpretable image transformations, and confirms this claim by a large number of qualitative examples. The authors also perform a detailed analysis of the transformations and categorize the PCA directions accordingly, showing that the first principal directions mostly affect geometry, while the successive directions typically control style.

Strengths: 1) The authors address an important problem, given the common ML trend to the unsupervised learning. The quality of sota GANs is excellent, and their usage in image editing applications is an important next step. 2) The provided evidence does demonstrate that StyleGAN contains directions that were not discovered by the previous methods. 3) The authors' approach is very simple, which is a virtue. 4) I really appreciate the analysis of different transformations (sect 3.1) and analysis of a distribution in the W space, a very interesting read.

Weaknesses: 1) The vast majority of the experiments is performed only for a single GAN model - StyleGAN, hence the authors' approach can lack generality. The directions discovered for BigGAN (zoom, translation) were reported e.g. in [8]. The "season" direction (Figure 7) is actually entangled with the object's appearance. Overall, the BigGAN experiments are not convincing enough, and method seems to rely heavily on the StyleGAN design. 2) It is not clear if interpretability comes from the PCA directions or from the layer-wise editing. Maybe, using the canonical directions with layer-wise editing would also result to interpretable manipulations. From my experience, many of canonical directions in both StyleGAN and BigGAN are interpretable, so, maybe PCA is not an essential component. Moreover, I do not agree with the authors' claim "L189: canonical directions in z are equivalent to random directions", since canonical directions in BigGAN are much more interpretable compared to random directions (at least from my experience). 3) Most of the results are qualitative, without any quantitative evaluation. Moreover, from the provided qualitative samples I would not say that the authors' method visually outperforms [8], which also does not require human supervision. I understand that measuring interpretability is challenging, but from the current set of results it is difficult to make reliable conclusions. 4) From the current version of the text, it is not clear how to determine the optimal subset of layers for layer-wise editing without exhaustive search, which can be quite time-consuming since it requires to inspect O(d*L^2) directions, where L is a number of layers in the generator. 5) I believe that several relevant prior works should be discussed: https://arxiv.org/pdf/1812.01161.pdf - this paper has shown that the eigen vectors of the generator's jacobian matrix typically correspond to interpretable directions. To the best of my knowledge, this is the first evidence that interpretability is related to the spectrum of the generator. Of coarse, the authors' method is more applicable since only the spectrum of the first layer should be computed, which is more scalable. Controlling generative models with continuos factors of variations, ICLR'2020 - also discovers simple interpretable directions without human supervision. Unsupervised Discovery of Interpretable Directions in the GAN Latent Space, ICML'2020 - a recent work that addresses the same problem.

Correctness: As I stated above, most of the experiments are qualitative and it is difficult to understand if the discovered directions are "pure"/disentangled.

Clarity: Yes, the paper is clearly written and easy to follow.

Relation to Prior Work: I am mostly satisfied with the related work section, but in the section above I have listed the additional relevant works to discuss.

Reproducibility: Yes

Additional Feedback: Given the weaknesses described above, I cannot recommend acceptance. While I like the overall premise and the simplicity of the approach, the reported findings are not general enough, since most of them are about StyleGAN, which design already has a bias towards disentanglement. ::::::::::::POST_REBUTTAL::::::::::: I have read the rebuttal, and I'm leaning to keep my original recommendation (4): 1) The authors have not addressed my question "It is not clear if interpretability comes from the PCA directions or from the layer-wise editing". There are no comparison to the random directions applied only to certain subset of layers in the submission, and the authors have ignored this concern in the rebuttal. This point is the most important, since maybe there is no need to perform PCA and using random/canonical directions with layer-wise editing will result in interpretable transformations. 2) In their experiments, the authors always compare to random directions from Z, instead of W space in StyleGAN, which is designed to be more disentangled. The important baseline of using canonical directions in the W space is missed. Overall, I am not convinced that the PCA directions are a necessary ingredient of the method, maybe one can just use the canonical W directions with layer-wise editing.


Review 2

Summary and Contributions: This paper studies interpretable controls for image synthesis such as change of viewpoint, aging, lighting, and time of day. First, important latent directions based on Principal Components Analysis (PCA) applied either in latent space or feature space. Then, authors show a large number of interpretable controls be defined by layer-wise perturbation along the principal directions. The proposed technique can be applied to any pretrained GAN model, such as BigGAN. Qualitative results shown on different datasets.

Strengths: The paper is first to show unsupervised identification of interpretable directions in an existing GAN. The paper shows that important directions in GAN latent spaces can be found by applying Principal Components Analysis (PCA) in latent space for StyleGAN, and feature space for BigGAN. Secondly, the authors show how BigGAN can be modified to allow StyleGAN-like layer-wise style mixing and control, without retraining. The method transfers PCA basis to latent space by linear regression. Authors enable Layer-wise Edits for BigGAN as well via intermediate latent vectors. The paper shares interesting findings in PCA components of BigGAN and StyleGAN.

Weaknesses: I do not have any issues with the paper and the paper provides an interesting way to explore GANs. The evaluation of the method is mostly qualitative. Do authors have any idea on how to compare different methods/component extraction? Another question is how does different datasets affect the components, such as camera motion and color transformations?

Correctness: Claims and methods seem to be correct. Authors confirm their claims with detailed visual experiments in the paper and sup. material.

Clarity: The paper is easy to understand and well written. Transaction between sections are good and I haven’t experienced many typos. Authors detail each section nicely and it is easy to follow.

Relation to Prior Work: The paper clearly explains different techniques in visualisation of GANs. In comparison, different supervised techniques are used for comparison and the proposed method achieves similar performance without any supervision. The paper is first to explore unsupervised discovery of Interpretable GAN Controls.

Reproducibility: Yes

Additional Feedback: Do authors have any idea on how to apply the proposed technique in the supervised case? ## Post Rebuttal Comments: I have read all reviews and the author's feedback. Reviewer 1 has several concerns about the use of PCA directions and it is concerning that the authors did not address this question clearly given that there were not many technical questions from the reviewers during the rebuttal. R1 also shares some interesting papers, which shows that PCA may not be necessary for this problem. Since there is no possible quantitative evaluation it is difficult to compare different techniques. Additionally, these papers published recently I have decided not to include them in my review. I will keep my rating the same as I believe this is a strong submission that investigates an important problem and the proposed solutions are simple and effective. I would strongly advise authors to include additional evaluations/comparisons for PCA vs. random/canonical directions with layer-wise editing.


Review 3

Summary and Contributions: This paper introduces conceptually simple way to discover interpretable controls for high-level image features (like camera, lighting, background, color and so on) of BigGAN. These major factors of variation function are very similar to StyleGAN styles and are discovered via PCA-based technique (unsupervised!). Thus it provides an interactive way to alter generated images without GAN retraining.

Strengths: I find it very interesting practical work that demonstrates how BigGAN decomposes images. The authors show that BigGAN uses latent space in a very similar way as StyleGAN without explicitly being trained so. They provide a GUI to study discovered controls interactively. Evaluation is done on FFHQ and ImageNet. Suggested PCA-based technique is simple to describe and also works on StyleGAN although not clear if it brings here besides already existing layer-level controls.

Weaknesses: I can't really think of any major problems with this paper, it is really great. The only thing perhaps is the role of StyleGAN demonstration, I found it quite confusing. I am not entirely sure why the authors inserted it: either to describe how they arrived to this idea (it is indeed a bit simpler on StyleGAN) or to demonstrate that it generalizes beyond a single flavour of GANs or to explain to an unfamiliar reader what controls it has. Either way I suggest to clarify it in the text.

Correctness: I think everything is correct here. Application of PCA makes sense and ultimately images look very convincing.

Clarity: The paper is very well-written and pleasure to read. It is somewhat unorthodox to have related work in the very end, but overall acceptable.

Relation to Prior Work: It looks like the major difference is unsupersived nature of their technique. They provide a comparison with a couple of supervised counterparts and a dumb baseline.

Reproducibility: Yes

Additional Feedback: POST REBUTTAL: I decided to keep my original evaluation despite concerns of other reviewers. I found the rebuttal satisfying.


Review 4

Summary and Contributions: This paper introduces a way of controlling the generation of a GAN for targeted high-level concepts via a latent space manipulation. The directions to move in latent space are learned in a relatively simple PCA-based way.

Strengths: -- Connecting the concepts of the skip-z in BigGAN to the StyleGAN (vis-a-vis controlling different levels of abstraction) is, to my best knowledge, a novel and interesting thought. -- The discussion of biases in the dataset is very interesting. Observing the inability to find latent directions that translate the image in the face datasets (which are aligned) but being able to in the other datasets is intuitive and interesting. This could be used in analysis of non-canonical datasets, as well, where these things are not known a priori. -- In the comparison to supervised methods, the proposed method holds up surprisingly well despite being unsupervised. -- Figure 6 is especially interesting. Very little work analyzes BigGAN-type models in a cross-class way, as opposed to looking at multiple classees but one at a time. -- The discussion of "impossible combinations", i.e. attribute co-dependencies, is very interesting. It is not a flaw that changing the type of car to "sporty" also changes the background and road, if the dataset has no "sporty" cars on the original background. This shows the GAN learning dependencies from the dataset, and could be used to learn about subtle relationships between high-level concepts in other non-canonical datasets, too.

Weaknesses: -- The authors merely state that performing this manipulation in layers after the first MLP-projection do not work well for the BigGAN, but showing this would be interesting. Those results may also provide insight into the underlying representations the GAN is learning. -- I'm skeptical that the analysis in Figure 6 would hold in many cases. The effect chosen is very forgiving in the sense that "adding green hue" can be done in a lot of ways without introducing bad artifacts in the image. I can't really trust this analysis until it's done on more challenging features like the pose of the foreground object or its shape. -- The comparisons to the supervised methods are a little unclear. Some of their directions are chosen and called "zoom", "glasses", "gender", etc... and compared to the corresponding supervised edit. How was the unsupervised direction (for glasses, for example) chosen? Were many different unsupervised directions considered, and then the one that looked like it corresponded to "glasses" was taken? How many were considered before finding it? This seems like a very manual, subjective, and tedious process. -- More investigation of which attributes don't have a direction to edit them that can be found in the unsupervised way would be beneficial. The discussion of tangled attributes (e.g. wrinkles on a baby's face) is related, but not directly addressing whether some attributes simply don't have corresponding directions in the latent space.

Correctness: No claims or methods appear incorrect.

Clarity: The paper is well-written and easy to follow. The message is targeted, all sections clearly advance that message, and there are no distracting tangents that too often muddy otherwise clear papers.

Relation to Prior Work: This work places itself in the literature adequately. I'm not aware of any methods that should have been compared to but weren't.

Reproducibility: Yes

Additional Feedback: Post Rebuttal: I have read all of the reviews and the authors' rebuttal to those reviews. Reviewer #1 makes a strong case that the manuscript as is doesn't sufficiently demonstrate that the PCA directions are beneficial, raising a new concern that I did not at first pick up on. I think this is an important point, as many claims are made about these directions being meaningful and I am now not convinced that they are. Moreover, my concerns from the original review about how the results and directions for comparisons are being picked, and the overall lack of strong quantitative evaluation, still remain. That being said, I find the layer-wise editing itself to be an interesting approach, whether the PCA directions are a substantial improvement over other directions or not. I also think it's valuable as it gives an interesting anecdotal insight into the internal representation of the BigGAN, as distinct from the contribution of the new method of editing. The strength of that aspect of the approach still has me leaning towards accept, but the previously mentioned concerns are significant enough drawbacks to prevent it from warranting a higher score.

[Author Response · NeurIPS 2020]

We thank the reviewers for their constructive and thorough feedback.

**Reviewer #1**

**1) BigGAN experiments not convincing:** Our submission demonstrates many directions for BigGAN that were not demonstrated by [8]: changing seasons, adding clouds, adding grass, day-night, warm lighting, pixelation, contrast, light direction, sharpness, owl height, background color, etc. (see Figure 7; Supplemental Material: Figure 1; and accompanying video: 4:00-4:40). While one can debate the merits of the entangling in Figure 7 (dogs photographed in snow may generally have heavier coats), we believe that these examples ought to be sufficient to convince the reader of the promise of our method for BigGAN. Moreover, we introduce style mixing for BigGAN, which is also novel.

**2) Canonical directions instead of PCA:** In our experiments with StyleGAN and BigGAN, we haven't been able to find canonical directions that were any more interpretable than random directions are (and random directions are sometimes somewhat interpretable). The published training algorithms do not give a specific role to canonical directions, making them no different from random directions. We are happy to mention this in the paper.

**3) Qualitative Evaluation:** It is true that some of our results are comparable to those of [8]. The main advantage of our method is that we can find many transformations that [8] cannot find, because that method requires hand-specified transformations as supervision. While we believe that the extensive demonstrations we provide in the paper, video, and supplement illustrate the promise of these ideas, we also believe that quantitative evaluation is useful. We are aware of no methods that would enable evaluation for large collections of interpretable directions, as demonstrated here, and think it is an extremely interesting direction for future work. We are happy to discuss this in the text.

**4) How to find layers:** Some effort is required to identify useful layer ranges. However, it does not require $L^2$ search; e.g., we find that certain ranges tend to be useful, and that there is no need to try arbitrary subsets. Moreover, we argue that this effort is far less than that of gathering supervised data, especially when one doesn't even know what attributes are controllable within a given GAN. See also the response to Reviewer #4 re "How was comparison performed".

**5) Prior and concurrent work:** Thank you, we will add and discuss these references in the revised paper. Ramesh (ICLR 2020) is indeed relevant in the way R1 mentions; the paper addresses a different problem from us. The method of Plumerault (ICLR 2020) seems very similar to [8], which we discuss and compare to. Please note that the ICML 2020 publication date was after the NeurIPS 2020 submission deadline.

**Reviewer #2:**

**How to evaluate quantitatively:** This is an interesting question; one possibility is to compare on a computer-generated dataset with known attributes. **How dataset affects components:** One observation we report in the paper is that translation is not discovered for StyleGAN faces, because FFHQ is already carefully aligned. This, together with the other entanglements we report, suggest that the components indeed are dataset-dependent. **Using this method in the supervised case:** One possibility is to linearly train on a small supervised dataset to use a sparse set of these PCA features. Another possibility, based on our layer-wise editing, is to learn a separate latent direction vector for each layer. We will mention this as future work.

**Reviewer #3:**

**Not clear if PCA helps on StyleGAN:** As shown by Figure 4 and the Supplemental Material (Figures 2–5), the PCA basis gives a useful content-style separation and ordering of directions. For example, all random directions seem to include some pose and appearance variation, whereas, in PCA, pose variations only occur in the first 20 or so components. **Role of StyleGAN demonstration:** We argue that the techniques we describe for StyleGAN are themselves useful, since we provide many ways to control StyleGAN models that have not been discovered before.

**Reviewer #4**

**— Using later layers for BigGAN:** We offer to add examples to the supplemental showing results using later layers of BigGAN.

**— Analysis in Figure 6:** See Figure 10 of the Supplement for more dramatic examples.

**— How was comparison to supervised methods performed:** Many of the comparisons were based on directions we'd already found, and some we found specifically for this comparison. For the latter, it took at most five minutes to pick a suitable component (often there were several good candidates to choose from) and choose the layer range.

The purposes of these examples is to show that some of the edits found by supervised methods also emerge in our technique. Some effort is required to sample our PCA directions and layers, but we argue that this is less than the effort of creating supervised data.

**— More investigation of which attributes can't be found:** This is an interesting avenue for future investigation.

[Meta-Review · NeurIPS 2020]

This paper studies interpretable controls for generative models using PCA directions in latent space. The proposed technique is natural and applied to Stylegan and BigGAN showing very good empirical results. Despite concerns by some reviewers that PCA directions are not necessarily the right thing to do, they are reasonable and work in practice, so we overall think this paper is a suitable contribution for NeurIPS. Please include the mentioned related works in the final manuscript.